# The Impact of EndoVAC in Addressing Post-Esophagectomy Anastomotic Leak in Esophageal Cancer Management

**DOI:** 10.3390/jcm13237113

**Published:** 2024-11-25

**Authors:** Stavros P. Papadakos, Alexandra Argyrou, Ioannis Katsaros, Vasileios Lekakis, Georgia Mpouga, Chrysovalantis Vergadis, Paraskevi Fytili, Andreas Koutsoumpas, Dimitrios Schizas

**Affiliations:** 1Department of Gastroenterology, National and Kapodistrian University of Athens, Laikon General Hospital, 115 27 Athens, Greece; stavrospapadakos@gmail.com (S.P.P.); argyalex89@gmail.com (A.A.); lekakis.vas@gmail.com (V.L.); gioulibouga@gmail.com (G.M.); info@drfytili.gr (P.F.); andreas.koutsoumpas@laiko.gr (A.K.); 2First Department of Surgery, National and Kapodistrian University of Athens, Laikon General Hospital, 115 27 Athens, Greece; 3Department of Radiology, Laikon General Hospital, 115 27 Athens, Greece; valvergadis@yahoo.gr

**Keywords:** EndoVAC, negative pressure wound therapies, NPWT, esophageal cancer, endoscopic therapy, esophagectomy, anastomotic leakage

## Abstract

Anastomotic leakage (AL) remains a major complication after esophagectomy, especially in patients with esophagogastric cancers who have undergone neoadjuvant therapies, which can impair tissue healing. Endoscopic vacuum-assisted closure (EndoVAC) is an innovative approach aimed at managing AL by facilitating wound drainage, reducing infection, and promoting granulation tissue formation, thus supporting effective healing. This review explores the role and effectiveness of EndoVAC in treating AL post-esophagectomy in esophageal cancer patients. We present an overview of its physiological principles, including wound contraction, enhanced tissue perfusion, and optimized microenvironment, which collectively accelerate wound closure. In addition, we examine clinical outcomes from recent studies, which indicate that EndoVAC is associated with improved leak resolution rates and potentially shorter hospital stays compared to traditional methods. Overall, this review highlights EndoVAC as a promising tool for AL management and underscores the need for continued investigation to refine its protocols and broaden its accessibility. By optimizing EndoVACs use, multidisciplinary teams can improve patient outcomes and advance esophageal cancer care.

## 1. Introduction

Malignancies of the esophagus and gastroesophageal junction (GEJ) constitute a highly diverse group of cancers with an extreme dichotomy in incidence among Western and Eastern populations [1,2], in their pathologic classification [3], the molecular pathways involved in their pathogenesis, as well as their therapeutic management [4,5]. Esophageal cancer (EC) comprises the eight most commonly diagnosed malignancies and the sixth most frequent cause of cancer-associated death [6]. According to GLOBOCAN reports, the incidence of esophageal carcinoma displays an upward trend [7,8]. In 2018, 572,000 new cases of EC with 508,000 deaths had been reported, with an increase in 604,000 new diagnoses and 544,000 deaths in 2020 [7,8]. An inclining surge in the esophageal adenocarcinoma (EAC) has been documented during the past decades in the developed world following the rising incidence of Barrett’s esophagus (BE) [1]. Although, until recently, considered to have a poor prognosis, several developments show promising trends across various aspects [9,10]. Firstly, findings from the International Cancer Benchmarking Partnership (ICBP-SURVMARK-2 project), comparing data from 1995–1999 to 2011–2014, reveal a nearly doubled 5-year survival rate for both EAC and esophageal squamous cell carcinoma (ESCC), particularly benefiting patients under 75 years old [11,12]. Secondly, in the current landscape where combined therapies supersede sole surgical interventions for treating locally advanced cases, the benchmark for 5-year survival now nears 50%, marking a significant increase over two decades [13,14]. Thirdly, enhanced detection rates of early-stage lesions in the mucosal and submucosal layers result from heightened cancer awareness, surveillance of BE, management of gastroesophageal reflux disease (GERD—a risk factor for BE and thus EC), and advancements in staging techniques. These improvements facilitate the relatively low-risk treatment of selected patients through endoscopic eradication therapies (EET), including procedures like endoscopic mucosal resection (EMR), endoscopic submucosal dissection (ESD), and radiofrequency ablation (RFA) [15]. According to the aforementioned information, despite the significant improvement in survival rates achieved through the surgical management of esophageal cancer, anastomotic leakage (AL) remains the most serious complication following esophagectomy, continuing to be associated with high mortality rates [10]. Various techniques have been employed to address these complications, yielding differing outcomes [15]. Among these, EndoVAC is an innovative endoscopic technique specifically designed to manage Als, providing a minimally invasive and effective solution.

### 1.1. The Esophagectomy Techniques

Esophagectomy comprises the only therapeutic intervention with curative intent in patients with locally confined disease. A plethora of esophagectomy techniques have been documented by the Society of Thoracic Surgeons [16], which are further classified as open, minimally invasive, and hybrid approaches. Transthoracic esophagectomies, namely Ivor Lewis, McKeown, and esophagectomy through a left thoracoabdominal incision, confer innately superior oncologic results due to the more extensive lymphadenectomy they offer [4], notwithstanding a severe complication profile including pneumonia, mediastinitis, and sepsis from AL [17]. Conversely, transhiatal esophagectomy demonstrates a more favorable complication profile by avoiding the potential occurrence of an anastomotic leak within the thoracic cavity, but this advantage comes at the cost of a less thorough lymph node dissection [4]. The stomach comprises the most frequently used conduit, with interposition of the colon or jejunum being secondary surrogates [18]. Gastric conduit, especially when anastomosed with the cervical esophagus, is more prone to ischemia due to the extensive tissue mobilization. That constitutes a major determinant for the development of AL [19]. It should be further highlighted that preoperative chemoradiation, when offered to patients with locally advanced disease, results in significant tissue ischemia, which also predisposes to anastomotic leak [19]. It becomes easily perceived that esophagectomy comprises an onerous procedure with significant morbidity and mortality [16,20], and the management of AL becomes a cornerstone in the therapeutic approach of esophageal and GEJ carcinoma [21].

### 1.2. Aim of the Article

The aim of this article was to provide a comprehensive review of the use of endoscopic vacuum-assisted closure in managing anastomotic leakage following esophagectomy in patients with esophageal cancer. This review explores the physiological principles underlying EndoVAC, assesses its clinical outcomes, and examines its advantages over conventional therapies, including reduced infection risk, enhanced granulation, and shortened recovery times. By analyzing current evidence and presenting emerging data on EndoVACs efficacy, the article aims to guide clinicians in its optimal application and encourage further research to refine protocols, improve accessibility, and ultimately enhance patient outcomes in esophageal surgery.

## 2. The Anastomotic Leakage in Esophagectomy for Malignancy

As mentioned above, AL is the most severe complication after esophagectomy [22]. Well-documented risk factors for the development of AL include the presence of cervical anastomosis, increased BMI, heart failure, coronary artery disease, vascular insufficiency, renal disease, and smoking [23]. Towards the same direction, Griffiths et al. aimed to identify preoperative risk factors for AL and conduit necrosis (CN) following oesophagectomy for EC. The research used data from the international Oesophago-Gastric Anastomotic Audit (OGAA) between April and December 2018, covering 2247 oesophagectomies from 137 hospitals across 41 countries. They showed an AL rate of 14.2% and a CN rate of 2.7%. For AL, cardiovascular comorbidity and chronic obstructive pulmonary disease were identified as independent predictors. In contrast, for CN, factors like BMI, Eastern Cooperative Oncology Group (ECOG) performance status, previous myocardial infarction, and smoking history were found to be independent predictors. A risk-scoring model based on these factors demonstrated better predictive ability with an area under the receiver-operating characteristic curve (AUROC) of 0.775 in internal validation [24].

Neoadjuvant therapy, involving chemotherapy (nCT) or chemoradiotherapy (nCRT) administered before surgical resection, is increasingly utilized in the management of EC due to its potential to downstage tumors and enhance surgical outcomes. Studies have shown that neoadjuvant therapy can significantly reduce tumor size and improve the resectability of the tumor, thereby increasing the likelihood of complete tumor removal during surgery. However, the use of neoadjuvant therapy also raises concerns regarding AL [25]. Research indicates that neoadjuvant therapy may increase the risk of AL compared to upfront surgery, primarily because of radiation and chemotherapy on tissue integrity. Radiation can cause fibrosis, decreased blood supply, and tissue necrosis, which compromise healing and increase the risk of leaks at the anesthesia site. Chemotherapy, while beneficial in reducing tumor burden, can also impair immune function and delay tissue repair, further contributing to the risk of leakage. Despite the increased risk of AL, neoadjuvant therapy offers significant benefits in terms of overall survival and disease-free survival. The survival advantage provided by neoadjuvant therapy often outweighs the risk of postoperative complications when managed appropriately [26]. Advanced surgical techniques and intraoperative measures, such as the use of reinforcing materials like fibrin glue and the implementation of intraoperative leak testing, have been developed to reduce the incidence and impact of AL [27]. Additionally, the involvement of multidisciplinary care teams ensures comprehensive perioperative care, addressing potential complications promptly and effectively [28]. In summary, while neoadjuvant therapy in esophageal cancer treatment may increase the risk of AL due to radiation-induced tissue damage and chemotherapy-related healing impairments, its overall benefits in tumor reduction and survival enhancement make it a valuable approach. Both meticulous surgical technique and perioperative management of the patients are essential to mitigate the risks and maximize the therapeutic advantages of neoadjuvant therapy in esophageal cancer.

The absence of a standard definition for AL led to the establishment of the Esophagectomy Complications Consensus Group (ECCG) [25], which defined AL as transmural dehiscence of the esophagus, anastomosis, or conduit and further classified them as types I, II, and III according to their severity and the necessity of intervention. Type I Als can be managed conservatively with nil per os, broad-spectrum antibiotics, antifungals, and close monitoring, while type II Als require interventional management and type III surgical revision [29]. Consequently, it is understandable that AL is categorized by ECCG according to their management and not on the basis of endoscopic and imaging characteristics [30]. According to the OGAA clinical study, which implemented the abovementioned definitions, the estimated rate of AL reaches 14.2%, and more specifically, types I, II, and III represent 7.0%, 3.4%, and 3.8%, respectively. Additionally, a more refined classification could be conducted by distinguishing between intrathoracic and cervical leakages. To this end, Kassis et al. conducted an analysis using data from the Society of Thoracic Surgeons Database, examining 7595 esophagectomies performed between 2001 and 2011, of which 804 (10.6%) resulted in leaks. Their findings indicated that the incidence of leaks was notably higher in patients with cervical anastomosis compared to those with intrathoracic anastomoses, at 12.3% versus 9.3%, respectively (*p* = 0.006) [23]. To substantiate these conclusions, a recent meta-analysis of 12 randomized controlled trials, involving 1493 patients, demonstrated a significant pooled incidence of AL (4.8% for intrathoracic anastomosis vs. 13.6% for cervical anastomosis) [RR = 2.76, 95% CI (1.94–3.94), *p* < 0.001]. This finding indicates a considerably higher risk in the cervical anastomosis group compared to the intrathoracic anastomosis group [31]. Furthermore, it highlighted that intrathoracic anastomosis is correlated with a notably diminished risk of a leak. A higher grade of AL was correlated with a longer duration of hospitalization, a greater mortality rate, and worse quality of life (QoL), as reflected by the deteriorated nutrition on discharge [31].

Ubels et al. conducted a case vignette analysis detailing the status and future prospects of endoscopic vacuum-assisted closure (EndoVAC) in managing AL [32]. They consulted upper gastrointestinal expert surgeons, many of whom were members of organizations like the International Society for Diseases of the Esophagus (ISDE), OGAA, and the TENTACLE—Esophagus study; chosen based on their extensive institutional experience (performing over 60 resections annually); personal expertise; and contributions to AL research. The primary objective of their interventions was to prevent mortality associated with AL, with secondary aims including promoting anastomotic healing, facilitating cautious discharge, improving quality of life, and reducing recontamination risk. Treatment strategies advocated by experts emphasized drainage of fluid cavities, closure of dehiscences, and adjunctive therapies, aligning with physiological principles. Non-surgical interventions were preferred for fluid collections, tailored according to the collection site, contamination severity, and local protocols. Pleural collections were considered more critical than mediastinal ones due to potential spontaneous drainage through dehiscences. Management decisions for conduit ischemia and multiorgan dysfunction were guided by conduit viability and systemic sepsis severity, respectively, with surgical revision reserved for non-salvageable conduits and cases of multiorgan failure. Experts categorized patients into groups based on disease extent (local or widespread) and anatomical considerations (pleural collections or conduit ischemia). Despite promising outcomes, EndoVAC utilization remains modest (approximately 64%), contrasting with higher availability rates for alternative modalities such as US- and CT-guided drainage, endoscopic stenting, clips, minimal invasive, and open surgical approaches, suggesting underutilization of EndoVAC despite its potential benefits [32].

Taking a step further, more recent evidence categorizes the therapeutic principles of AL according to their physiological mechanisms into three subgroups: (a) the puncture/aspiration of fluid collections; (b) the closure of the anastomotic dehiscence with various techniques such as stent installation, surgical revision, or esophageal diversion; and (c) supportive approaches with antimicrobials and feeding continuation. The overall treatment approach could be constituted by various combinations of the above principles with a multitude of endoscopic, interventional, or surgical techniques [32].

## 3. The Role of EndoVAC in the Management of EC Anastomotic Leakage

### 3.1. Physiologic Principles of EndoVAC Action

The underlying physiologic principles of EndoVAC and negative pressure wound therapies (NPWT) have been exhaustively reviewed [33,34], and their detailed reference goes beyond the scope of this manuscript. Briefly, Huang et al. have proposed the principles of EndoVAC wound healing, encompassing the reduction of wound surface area, or “macrodeformation”, the creation of a corrugated wound surface through applied negative pressure, or “microdeformation”, the evacuation of extracellular fluids, and the optimization of the wound microenvironment. Several advantages are derived from the above EndoVAC effects [35]. While macrodeformation results in wound contraction, microdeformation induces cellular proliferation and differentiation, transmitting the contraction forces through the extracellular matrix stroma [36]. The draining of fluids exerts its primary effects by improving tissue perfusion, enhancing the circulation in microvasculature, removing contaminants such as exudate, toxins, and microorganisms, and augmenting the lymphatic draining from the wound edges [37,38,39]. Finally, foam materials are utilized in order to interfere with further contamination of the wound bed with bacteria [40] and to maintain the proper humidity [35].

Consequently, a series of secondary biological events are triggered, exerting an impact in the wound healing stages, namely: hemostasis, inflammation, proliferation, and remodeling phases [41]. Besides hemostasis, which should be completed upon the EndoVAC application, especially in patients with coagulation disorders, the other wound healing processes are positively influenced [35]. Nuutila et al., in a clinical study comparing the effects of NPWT application in injuries resulting from contusion, thermal damage, ischemia, and thrombolysis, documented a significant upregulation in the expression of genes involved in inflammatory response and cellular proliferation [42]. The IL24 gene has documented a nearly 10-fold increase in its expression pattern, with PTGS2 demonstrating an 8-fold increase. Altogether, several interleukins with their receptors (such as IL1RL1, IL8, IL1A, and IL13RA2), prostaglandins, chemokines, and matrix metalloproteinases (MMPs) are upregulated in response to NPWT treatments [42,43]. Wilker et al. demonstrated in vitro employing 3D fibroblast cultures that the application of NPWT resulted in cells with bulkier cellular bodies and enhanced proliferative capacity [44]. As regards the angiogenesis [45] and the neurogenesis [46], both processes are stimulated during EndoVAC treatment. The microdeformation establishes a hypoxia gradient and consequently a VEGF gradient, triggering the hypoxia-inducible factor-1α (HIF-1a) [45], while an upregulation in the expression of neuropeptides (such as substance P, neurotrophin, epinephrine, and norepinephrine) is reported [46]. Summarizing the above preclinical data, it becomes easily perceived that negative pressure therapies specifically EndoVAC exert their multilayer beneficial effects at molecular and cellular level.

### 3.2. EndoVAC Pioneers

The integration of EndoVAC into the management of AL following gastrointestinal (GI) surgeries has evolved through collaborative efforts among specialized gastrointestinal surgeons, interventional gastroenterologists, and radiologists. Weidenhagen et al. introduced EndoVAC in 2004, adapting vacuum therapy techniques from external wound management to rectal cancer surgery, marking its initial application in the context of AL [47]. Subsequently, Weidenhagen et al. reported on a case series involving 29 patients who underwent intracavitary EndoVAC following low anterior resection, achieving a notable success rate of 86.2% in achieving clinical healing without the need for surgical revision [48]. This early evidence highlighted EndoVAC as a promising alternative to conventional treatments.

In another pivotal study, Wedemeyer et al. demonstrated the efficacy of EndoVAC in two patients resistant to standard interventions post-esophagectomy and gastrectomy, where alternative measures such as surgical revision and self-expanding metal stents (SEMS) had failed [49]. This success spurred widespread interest and initiated numerous clinical studies across institutions, aiming to validate and refine the applicability of EndoVAC in diverse clinical settings [50,51].

The indications for employing EndoVAC therapy in the management of AL following esophagectomy are multifaceted and cater to scenarios where traditional interventions prove inadequate or pose heightened risks. EndoVAC therapy is particularly valuable in cases of persistent or recurrent leaks despite conventional treatments, offering a non-surgical approach to enhance wound healing and mitigate infection risks [32]. High-risk patient profiles, characterized by compromised tissue healing due to factors such as malnutrition or advanced age, further underscore its utility in improving clinical outcomes [32].

Additionally, EndoVAC serves as an adjunctive therapy alongside surgical interventions, facilitating wound closure and promoting tissue granulation in compromised tissue viability or poor wound perfusion scenarios [32]. Its application extends to managing anatomical complexities such as complex fistulas or strictures, aiding in tissue approximation and defect sealing [32,52]. Recent literature reviews, such as that by Virgilio et al., highlight its efficacy in managing localized leaks by reducing infection risks and accelerating wound healing, albeit with recognized limitations in treating extensive or multifocal leaks requiring multiple endoscopic sessions [52].

Recent studies have revisited earlier reservations regarding the use of EndoVAC in challenging scenarios such as intrathoracic leakage, conduit ischemia, mediastinitis, and systemic sepsis, suggesting its efficacy in reducing mortality rates and promoting successful outcomes [50,53,54,55,56,57,58]. Advances in managing large, chronic cavities exceeding conventional size limits have also expanded its clinical applicability, with studies reporting clinical success rates exceeding 90% despite initial concerns [59,60,61].

In conclusion, EndoVACs evolving role in AL management underscores the importance of tailored therapeutic approaches and interdisciplinary collaboration to optimize patient outcomes in complex GI surgery settings.

### 3.3. The EndoVAC Technique and Variations

The procedure of EndoVAC placement has been described by various groups [62,63,64,65]. An initial endoscopic examination of the anastomotic area in order to evaluate the size of the defect and the contiguous cavity has to precede. Additionally, the initial inspection aims to highlight structures within the cavity or areas difficult to drain. Its proximity to large vessels is assessed with computed tomography (CT) [63]. The index endoscopy should be carried out in a short period of time without debridement or irrigation of tissues, ideally insufflating CO_2_ instead of O_2_ in order to avoid air embolism or further dehiscence with endoscopes of up to 10 mm in width [65]. The size of the defect provides clinically useful information guiding the decision for intraluminal or intracavitary vacuum therapy [66].

A nasogastric tube (NGT) is placed through the patient’s nostrils and is extracted by the mouth while a specifically cut polyurethane sponge is sewn on its tip with Polydioxanone Sutures. Two separated holes (0.5 cm each) remain at the nasogastric tube near its tip and are covered by the sponge. The cavity should be slightly larger than the sponge in order for negative pressure to be constituted [67]. A suture loop should be constructed in the distal tip of the sponge to reduce the difficulty of grasping it through the upper esophageal sphincter and the insertion in the cavity [65], a method widely known as “piggyback” [68]. Manipulations for the placement of sponge could also be carried out at the tip of the NGT with endoscopic forceps [64]. Pines et al. described an alternative technique when the sponge is placed into the esophageal cavity through an existent tract in the pleural cavity, namely the “rendezvous technique” [53]. When the sponge is in place, a continuous negative pressure of 100 mmHg is applied, while changes should be carried out every 3 to 5 days until the entire healing of the cavity is achieved [65].

The EndoVAC application protects the cavity from exposure to highly acidic gastric juice which could be erosive, while as mentioned above it removes the microbial burden and the excess of the interstitial fluids. A variation among several authors has been documented with others proposing wider intervals for the changes (Lee et al. conducted changes every 4 to 7 days [64]) and others administering lower levels of intracavitary pressure (Gutschow et al. conducted EndoVAC therapy with 75 mmHg [65]). Loske et al. applying continuous negative pressure of 125 mmHg reported sufficient rates of wound sealing and drainage [66]. A recent investigation utilizing a porcine model for esophagectomy demonstrated that employing EndoVAC therapy at a pressure of −125 mmHg markedly augmented tissue perfusion in ischemic gastric conduits. This finding suggests a potential preemptive application of EndoVAC therapy for gastric conduits exhibiting compromised arterial perfusion or venous congestion [69]. The above are presented in a clinical case illustration in Figure 1.

While the EndoVAC alone could address the aspiration of fluid collections, recent combinatorial approaches have been described to encounter more complex ALs. Gubler et al. described the stent-over-sponge (SOS) technique where SEMS were utilized to better seal of an intracavitary-placed polyurethane sponge [70]. This procedure may expose patients to SEMS-related complications such as stent dislocation or complications following the stent removal, but if it succeeds, it prevents revision surgery, which is accompanied by a more severe complication profile [70]. The efficacy of this approach has been evaluated in a clinical study by Valli et al., where the success rate was 75% with no serious adverse events reported [71]. Lima et al. reported an alternative approach to deliver negative pressure by inserting a 12-Fr Levin into a wider 20-Fr tube to suction a mediastinal abscess [72]. The internal tube was connected with the aspiration device while the outer tube averted clogging [72]. Saraiva et al. documented the sponge installment under guidewire guidance [68]. The commercially accessible polyurethane sponges differ in porosity and density, which influence their insertion ability, shrinkage potential, and granulation tissue-formation capacity. On the other hand, open-pore film drains are more easily applicable with less tissue attachment, rendering them also more permissible for removal [63].

### 3.4. Results and Risk Factors for Failure

Despite its proven efficacy, EndoVAC remains underutilized, with availability reported at approximately 64% of centers performing esophagectomies, indicating ongoing challenges in its adoption compared to other established treatment modalities, such as self-expanding metal stents [32]. Studies (Table 1) have underscored the critical role of experience and familiarity with EndoVAC techniques in achieving successful outcomes. Interdisciplinary collaboration between surgical and gastroenterological specialties is essential for tailored patient management strategies and improved therapeutic efficacy.

Pattynama et al. reported an initial experience with EndoVAC for AL following upper GI surgery, documenting a 74% success rate and a 5% adverse event rate [78]. They analyzed 38 patients treated with EVT between 2018 and 2021. They measured the primary outcomes of EVT success in healing the leakage, with secondary outcomes including mortality and adverse event rates. They found that EVT achieved defect closure in 74% of cases, requiring an average of six EVT-related procedures over 33 days per successful treatment. For patients where EVT failed (26%), alternative treatments, including surgery, were necessary. They also examined adverse events (AEs) related to EVT, noting an AE rate of 5%, including two cases of severe complications. Mortality was reported at 8%, with deaths attributed to unrelated conditions such as radiation pneumonitis and pulmonary embolism. Despite a few complications, the authors concluded that EVT is a promising treatment for anastomotic leakage that could reduce the need for further major surgeries. They suggest that the efficacy of EVT may improve as providers gain more experience with the technique. Menningen et al. retrospectively analyzed the outcomes of 45 patients treated at their department for post-esophagectomy AL with a 93.3% success rate of EndoVAC treatment compared to 63.3% for metal stents [75]. Specifically, they investigated the effectiveness of EVT compared to stent placement for treating AL following esophagectomy. This single-center, retrospective study examined outcomes for 45 patients treated from 2009 to 2015, with 30 initially receiving stents and 15 managed with EVT. Notably, seven patients in the stent group were subsequently switched to EVT due to insufficient response, while four required surgery. They showed EVT had a significantly higher initial success rate (93.3%) for healing compared to stent placement (63.3%) with *p* = 0.038. After accounting for patients who changed therapies, EVT maintained a higher success rate (86.4%) versus 60.9% for stents, though this difference was not statistically significant (*p* = 0.091). Mortality rates, treatment duration, and hospital stay lengths were similar across both groups. These results suggest EVT could be a promising primary option for managing anastomotic leaks post-esophagectomy.

Furthermore, Brangewitz et al. reported an 84.4% success rate of EndoVAC (vs. 53.8% for metal stents) for intrathoracic AL post-esophagectomy, but no difference was found in total length of hospitalization or mortality compared to metal stent placement [73]. Schniewind et al. retrospectively analyzed 62 critically ill patients with post-esophagectomy AL, with EndoVAC patients having the lowest mortality rates [74]. In more detail, they evaluated various treatments for AL post-esophagectomy by comparing EVT, surgical revision, stent application, and conservative management. Of 366 patients who underwent esophagectomy, 62 developed leaks; these were managed based on their APACHE II severity scores. Findings indicated that EVT significantly reduced mortality in critically ill patients compared to surgery and stent placement, with mortality rates of 12%, 50%, and 83%, respectively, for matched patients. Conservatively treated patients with mild conditions (APACHE II mean score of 5) had no mortality. This study highlighted EVTs effectiveness in managing severe anastomotic leaks, advocating for its efficacy as a primary treatment option.

Scognamiglio et al. systematically reviewed and analyzed five retrospective studies to compare the effectiveness of EVT and SEMS for treating postoperative esophago-enteric anastomotic leaks [73,74,75,76,77]. They showed that EVT was significantly more effective in achieving leak closure (odds ratio [OR] 3.14) and had a shorter treatment duration, with an average reduction of 11.9 days compared to SEMS. EVT also had a lower in-hospital mortality rate (OR 0.39) and required more frequent device changes. However, there were no significant differences in overall hospitalization time or major complications. Despite EVTs potential advantages, the study’s retrospective design warrants caution, and further prospective research is needed to confirm these findings and optimize treatment strategies for this complication [79]. These findings were further supported by a meta-analysis of 163 patients, where EndoVAC therapy had a higher closure rate (OR: 5.51, *p*-value <0.001), approximately 9 days shorter length of hospitalization, and significantly lower major morbidity rates compared to metal stents (OR: 0.38, *p*-value: 0.011) [80]. Reimer et al. explored the use of EVT in managing large AL (>30 mm) after upper GI surgeries. In this study, 92 patients with upper GI tract leakages were analyzed, with 20 having large defects. Patients with larger defects required significantly more treatment days (42 vs. 14) and hospital stay (63 vs. 26 days), as well as developed more septic complications (40% vs. 17.6%). However, the resolution of leaks in these patients was 80%, comparable to the 90% resolution in the control group. The study highlighted several technical challenges encountered during EVT for larger defects, including multiple defects, foreign material, and limited access to the defect site. Solutions such as using multiple sponge systems, necrosectomy, and improving access routes were employed successfully. Despite these challenges, EVT remains a viable option for treating large GI leaks, though it requires skilled management and may involve additional procedures to address complications [61].

While the results are promising, procedural details and patient-specific factors may significantly influence overall outcomes and the clinical course [78,81]. Seika et al. analyzed patients treated with EndoVAC following post-esophagectomy AL and concluded that neoadjuvant chemoradiation led to a significantly longer duration of treatment by approximately 6 days compared to neoadjuvant chemotherapy, but the success rate and mortality rates did not differ between the two groups [82]. Pattynama et al. examined retrospectively 27 patients with post-esophagectomy AL treated with EndoVAC and showed that patients with unsuccessful treatment had higher levels of CRP at AL diagnosis (300 vs. 228 mg/dL) and had larger defects (>40% of anastomosis circumference), but their results did not reach statistical significance mainly due to the small number of included patients [78]. Book et al. showed that intraluminal placement of EndoVAC had higher success rates compared to intracavity placement (success rate: 86% vs. 70%, *p*-value: 0.031) [81]. Furthermore, the development of complications due to EndoVAC, such as bleeding, EndoVAC dislocation, and tracheoesophageal fistula, significantly predisposed to treatment failure (success rate 84% vs. 54%, *p*-value < 0.001). Diabetes mellitus and malnutrition status did not significantly differ between the two groups and did not affect patients’ outcomes [83]. On the contrary, Jung et al. reported that intraluminal placement of EndoVAC had significantly worse outcomes compared to the intracavity placement (success rate: 38.3% vs. 61.7%, *p*-value: 0.014). CRP levels at diagnosis were significantly lower in the successfully treated group (40.8 vs. 74.2 mg/dl, *p*-value: 0.026). Neoadjuvant treatment was also associated with worse results (OR: 5.74, *p*-value: 0.002) [83].

Collectively, the utilization of EVAC over other techniques for managing esophageal anastomotic leaks is supported by solid evidence from a systematic review and network meta-analysis (NMA) of 12 retrospective cohort studies, involving 511 patients [84]. This study compared four interventions—stenting; EVAC; surgery; and conservative management—assessing key outcomes such as success; mortality; complications; further interventions; and hospital stays. EVAC demonstrated the highest success rate (78.2%) compared to other treatments, with surgery having a slightly higher success rate (79.5%) but also significantly higher mortality (28.1%) and larger defect sizes. Conservative therapy, while effective for smaller leaks, had the lowest mortality (1.4%) but did not show significant differences in other outcomes compared to stenting. In terms of complications, EVAC had the lowest rate (18.2%), significantly lower than stenting (39.8%), indicating it may offer a safer alternative. However, EVAC also had the longest hospital stay (51.9 days) compared to surgery (38.9 days) and conservative therapy (35.9 days), reflecting the more intensive nature of the treatment. Despite this, EVACs benefits in reducing mortality and complications support its use. Overall, EVAC is an effective option for esophageal AL, especially when compared to stenting and surgery. It offers lower mortality, fewer complications, and higher success rates, making it a preferable choice for many patients. Further research is needed to refine these findings and explore combination therapies.

### 3.5. Vacuum-Stent: Combining Endoscopic Vacuum Therapy and Intraluminal Stent

The VACStent, a fully covered self-expandable metal stent coated with polyurethane foam, was first utilized to treat an esophageal anastomotic leak in a 61-year-old male patient on the 16th postoperative day after a failed over-the-scope clip treatment [85]. The VACStent device consists of a 30 mm nitinol SEMS coated with silicone and a polyurethane sponge cylinder (10 mm thick) affixed to its exterior. The sponge connects to a vacuum pump via a 12F catheter. Negative pressure ranging from −50 mmHg to −120 mmHg is applied during treatment. The device is inserted transorally under fluoroscopic or endoscopic guidance, and correct placement is monitored using contrast fluoroscopy or endoscopy [86]. The VACStent provides a dual approach: the sponge applies negative pressure to promote wound healing while the SEMS ensures the GI passage remains patent for nutrition [85]. In its pilot study, three patients with upper GI leaks were treated using the VACStent. It facilitated wound healing via negative pressure while maintaining intestinal patency, allowing oral nutrition. No stent migration was observed, and all patients showed successful closure of their leaks [86]. Chon et al. (2021) conducted a study involving 10 patients treated with the VACStent^®^ hybrid SEMS for upper gastrointestinal leaks, achieving a 100% technical success rate across 15 interventions. The overall clinical success rate was 70%, with 80% success in first-line and 60% in second-line treatments, and no severe adverse events were reported [85].

These promising results have been further evaluated in prospective studies to better assess the efficacy and safety of this novel approach [87,88]. Chon et al. conducted a single-center study between September 2019 and November 2020, investigating the use of the VACStent for treating esophageal leaks. They involved 20 patients with 24 esophageal leaks, including 17 anastomotic leaks (70.8%), 4 iatrogenic leaks (16.7%), and 3 spontaneous leaks (12.5%). The VACStent was successfully placed in all 24 cases, achieving 100% technical success. Clinical success, defined as leak closure, occurred in 60% of cases (14/24). The median time to leak closure was 12 days (range: 7–21 days). The device was well tolerated, with no severe complications, although oral feeding failed in 4 cases due to stent obstruction. This study suggests that while the VACStent^®^ is a safe and feasible option for managing esophageal leaks, its superiority over conventional treatments like SEMS or EVT needs further investigation [87]. Lange et al. investigated the use of VACStent treatment for upper gastrointestinal leaks and anastomotic failures in 15 patients, demonstrating high technical success in covering leaks and promoting healing. In all 41 VACStent applications, leaks were successfully covered and drainage was facilitated through the PU-sponge cylinder, with 80% of patients achieving complete morphological healing within an average of 15 days. Most patients (87%) could resume oral intake, with 53% able to eat solid food. Despite minor complications like migration (7%) and tissue ingrowth in some cases, no major adverse events such as erosion, ulceration, or significant bleeding occurred. The treatment effectively controlled septic conditions, and although two patients required surgery for persistent leaks, the VACStent proved a safe and effective solution for managing complex gastrointestinal leaks, especially in critical care settings. Finally, Lange et al. (2023) conducted a study involving 50 patients who received 92 VACStent applications [88]. The patients had a mean age of 63.5 years, with 64% having an ASA score of 3. The majority of cases involved upper GI leaks, including postoperative anastomotic leaks and iatrogenic perforations. The VACStent was successfully placed in all cases, with a mean indwelling time of 5.2 days (range 2–8 days). A 70% healing rate (28/40 patients) was observed, increasing to 80% with additional treatments. In preemptive cases, 12 applications resulted in 100% anastomosis healing. Minor local bleeding occurred in 5.4% of patients, but no significant complications were reported. These results suggest that VACStent is a promising treatment for upper GI leaks, though larger studies are needed for further validation [89]. Finally, Lange et al. explored the preemptive use of the VACStent in high-risk patients undergoing hybrid esophagectomy after neoadjuvant therapy. The VACStent was applied to prevent AL and its complications. The results showed successful device application in all patients, no in-hospital mortality, and uneventful anastomotic healing without septic episodes or severe adverse events. The approach demonstrates the potential of preemptive VACStent use to improve outcomes in esophageal surgery and reduce postoperative morbidity, warranting further validation in larger studies [90].

In conclusion, the VACStent shows high technical success and promising clinical outcomes for treating upper gastrointestinal leaks, including anastomotic leaks, with effective wound healing and maintained GI patency. However, further large-scale studies are needed to confirm its superiority over conventional treatments.

## 4. Discussion

Surgery is the only treatment with curative intent in patients with localized esophageal cancer [91]. Recently, several approaches, such as neoadjuvant immunotherapy plus chemotherapy, have translated into clinical practice [92], while the surgical technique has evolved exceptionally [33,93,94]. Despite the advances in the surgical management of EC, the incidence of AL remains unaffected, indicating that the management of AL could influence greatly the clinical outcome of the disease [95]. Utilization of SEMS, or self-expandable plastic stents (SEPS), although it is most commonly preferred [32] due to fewer technical demands, comes with a more unfavorable complication profile [95]. EndoVAC application is the most effective treatment option and has been shown to have better results than stents [79,80].

Prompt diagnosis and immediate management of suspected AL are both crucial for the successful management of AL. The Oesophago-Gastric Anastomosis Study Group has demonstrated that non-operative management, such as EndoVAC, stent, radiological drain, etc., is favorable as the primary approach to anastomotic leakages following esophagectomy [29]. Furthermore, they reported that non-operative reintervention after failed primary management is often successful and should be preferred over surgical approaches [29]. Reimer et al. emphasized that upper GI endoscopy within 6 h after AL suspicion, followed by the immediate application of EndoVAC, significantly increases the likelihood of AL closure and successful discharge of the patient [61].

Based on our center’s extensive experience with AL management after esophagectomy, we advocate for the routine and immediate use of EndoVAC in all suspected cases of AL, without delay or waiting to classify the leak as type I or II. The traditional approach of watchful waiting or classifying AL before determining management can unnecessarily delay definitive treatment, increasing morbidity and potentially compromising patient outcomes. AL is a time-sensitive complication where early and aggressive intervention can significantly influence recovery, patient survival, and overall quality of life [61]. Implementing EndoVAC as the first-line treatment, regardless of the leak type, allows for consistent and effective control of sepsis, containment of the leak, and promotion of tissue healing [79]. This proactive approach ensures rapid decompression of the infected space, minimizes the risk of further contamination, and enhances the patient’s chance of a full recovery [69,78]. By employing EndoVAC in all suspected AL cases, we reduce the risk of delay, enhance consistency in care, and improve overall survival and patient outcomes, thereby making it the gold standard for AL management in esophagectomy.

The economic burden of EVT in the treatment of anastomotic leaks and gastrointestinal perforations is an important consideration, with considerable variation in costs across different healthcare systems. Research conducted by Baltin et al. (2019) highlighted that, in the German healthcare system, EVT for managing esophageal anastomotic leaks following esophagectomy incurs significantly higher costs than SEMS treatment [96]. Specifically, EVT resulted in an average deficit of €9282 per case, almost double that of SEMS, which amounted to a deficit of €5156. These differences are largely due to the extended hospital stays required for EVT and the insufficient reimbursement rates. Similarly, Ward et al. (2019) assessed the cost-effectiveness of EVAC for esophageal and gastric leaks, demonstrating that performing the procedure in a GI lab, as opposed to the operating room, can reduce the total cost by approximately 2.5 times [97]. Our experience with a homemade EndoVAC system has shown that it can be implemented at a total cost of less than €500 without compromising efficacy, as supported by other studies on homemade EVT systems [98]. Further studies are necessary to better understand the cost implications and reimbursement challenges of EVT, particularly in varied healthcare settings, in order to make the procedure financially sustainable.

EndoVAC is increasingly being explored for applications beyond its traditional use in managing anastomotic leaks, with promising outcomes in both preemptive settings and life-threatening conditions [99]. In preemptive use, studies such as the pilot trial of the VACStent have demonstrated its potential to reduce anastomotic leakage after high-risk esophagectomy. By deploying the VACStent proactively in patients with high-risk anastomoses, the procedure was shown to reduce postoperative complications and promote uneventful healing without significant adverse events. This preemptive approach suggests that the use of EndoVAC could help prevent life-threatening situations before they arise, though larger clinical trials are needed to validate these findings [90]. Furthermore, EndoVAC has proven effective as a rescue therapy in life-threatening conditions, such as esophageal perforations, where traditional treatment options have failed [100]. In these cases, EndoVAC provides a viable alternative by promoting closure of the perforation and mitigating further complications, which has led to positive outcomes even in critically ill patients. These expanding applications highlight the versatility of EndoVAC, making it a valuable tool in both preventive and emergency therapeutic strategies, while further studies are required to confirm its broader utility in clinical practice.

Several limitations prevent us from reaching out with robust conclusions. Firstly, the availability of the technique is relatively limited [32] and underutilized. That is responsible for the generation of selection bias, which reduces the value and effectiveness of the conducted studies. Secondly, despite the extensive literature research, the vast majority of studies document both benign and malignant causes of esophageal dehiscence (injuries, trauma, Boerhaave syndrome, AL from bariatric surgery, etc.) with huge inhomogeneity, which renders the comparison misleading. Thirdly, the utilization of the technique to the greatest extent by experts in tertiary medical centers may withhold evidence of its applicability in everyday clinical practice.

## 5. Conclusions—Future Perspectives

In conclusion, the EndoVAC system offers a promising solution for managing anastomotic leaks (AL) following esophagectomy, showing better outcomes compared to other non-operative methods, such as stents. By creating an optimized environment for tissue healing through the use of negative pressure, EndoVAC facilitates fluid drainage, reduces infection risks, and promotes granulation tissue formation. Studies have demonstrated that EndoVAC achieves superior success rates in leak closure, shorter hospital stays, and reduced mortality, making it an increasingly valuable tool in managing this life-threatening complication.

Future perspectives should focus on further refining EndoVAC protocols, including patient selection criteria, timing, and duration of therapy, to ensure its optimal application. Comparative research is also needed to clarify how EndoVAC compares to or complements other treatment modalities, particularly in complex cases like gastric conduit ischemia. Moreover, interdisciplinary collaboration between surgeons, gastroenterologists, and radiologists will be essential in developing a more comprehensive classification system for AL, not based on the management approach but rather on the physiological and anatomical characteristics of the leak. This could significantly advance personalized treatment strategies and improve outcomes for patients undergoing esophageal cancer surgery.

## Figures and Tables

**Figure 1 jcm-13-07113-f001:**
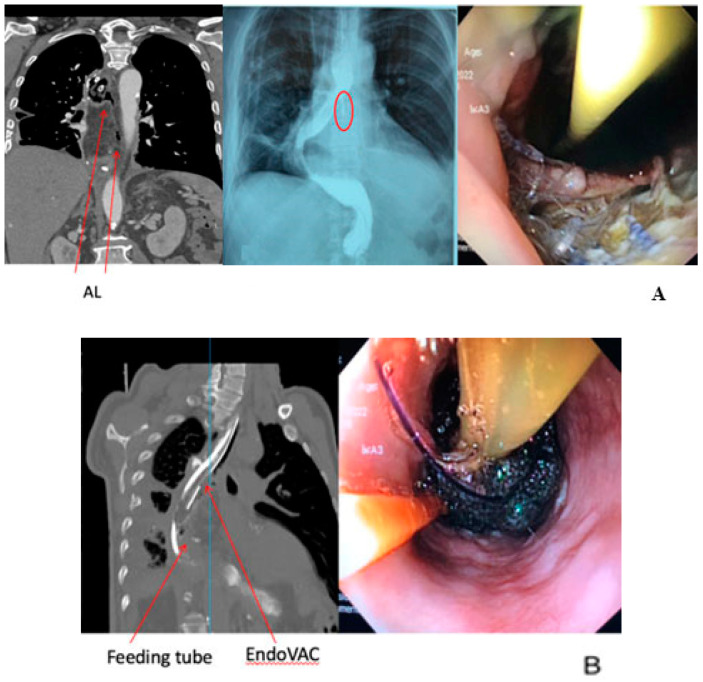
A 56-year-old male patient underwent an Ivor-Lewis esophagectomy post neo-adjuvant chemoradiotherapy (CRT). The surgical procedure included a hand-sewn esophagogastric anastomosis. (**A**) An AL was diagnosed on the 7th postoperative day. Arrows and circle show the AL. (**B**) Successful closure of AL was achieved after 27 days with 8 sessions of endoscopic vacuum therapy.

**Table 1 jcm-13-07113-t001:** Main studies comparing endoscopic vacuum therapy (EVT) to other treatment modalities of Anastomotic Leakage after esophagectomy.

Study ID	Patients	Main Findings
Brangewitz 2013 [73]	81 (32 EVT, 39 SEMS)	closure rate was significantly higher in the EVT group (84.4%) compared with the SEMS group (53.8%)
Schniewind 2013 [57]	47 (17 EVT, 12 SEMS, 18 surgery)	EVT patients had lower mortality (12%) compared to surgically treated (50%) cases and SEMS (83%)
Menningen 2015 [74]	45 (15 EVT, 30 SEMS)	success rate was higher for endoscopic vacuum therapy (EVT 93.3% vs. stent 63.3%)
Hwang 2016 [75]	18 (7 EVT, 11 SEMS)	EVT had 100% success rate compared to 63.6% for SEMS
Berlth 2019 [76]	111 (35 EVT, 76 SEMS)	SEMS and EVT show comparable results
El-Sourani 2022 [77]	20 (13 EVT, 7 SEMS)	EVT achieved closure in 92.3% and no EVT-related complications. SEMS therapy was successful in 85.7% with 28.6% procedure-related complications.

EVT: endoscopic vacuum therapy (EVT); SEMS: self-expanding metal stent.

## Data Availability

Not applicable.

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
