# Peer review of "The Impact of EndoVAC in Addressing Post-Esophagectomy Anastomotic Leak in Esophageal Cancer Management"

_jcm, 2024, doi:10.3390/jcm13237113_

Round 1
Reviewer 1 Report
Comments and Suggestions for Authors
Dear authors,
you submitted a narrative review on EVT for AL after esophagectomy. However, the paper does not go into great detail although EVT is considered the gold-standard for AL. Important meta-analysis are not included. Satements like "In conclusion, while ongoing research addresses gaps in knowledge and technique refinement, the current evidence supports the use of endoVAC in contained cavities smaller than 8cm and clean, unloculated cavities" are just incorrect. EVT can be safely used for extraluminal and large cavities and are in most instances the only option to adress such leaks. Furthermore, any references supporting this conclusion are completely missing.
Unfortunately, most of the points are a summary of results of other studies inthe field with no clear direction and the reader is left with any possible treatment algorithms.
A table summarizing relevant studies in the field are completely missing.
Comments on the Quality of English LanguageGrammar and spelling errors throughout the paper.
Author Response
Comment 1: you submitted a narrative review on EVT for AL after esophagectomy. However, the paper does not go into great detail although EVT is considered the gold-standard for AL. Important meta-analysis are not included.
Response 1: Thank you for this valuable comment. We revised the manuscript to include additional details on the use of EVT for anastomotic leaks following esophagectomy. Relevant meta-analyses and comprehensive data supporting EVT as the gold standard for managing anastomotic leakage are now included in the revised manuscript.
Comment 2: Satements like "In conclusion, while ongoing research addresses gaps in knowledge and technique refinement, the current evidence supports the use of endoVAC in contained cavities smaller than 8cm and clean, unloculated cavities" are just incorrect. EVT can be safely used for extraluminal and large cavities and are in most instances the only option to adress such leaks. Furthermore, any references supporting this conclusion are completely missing.
Unfortunately, most of the points are a summary of results of other studies inthe field with no clear direction and the reader is left with any possible treatment algorithms.
A table summarizing relevant studies in the field are completely missing.
Response 2: Thank you for your input. We revised the conclusion to reflect the broader applicability of EVT. We have included a new table (table 1) summarizing key studies comparing EVT to other treatment modalities in anastomotic leak management.
Reviewer 2 Report
Comments and Suggestions for Authors
S Papadakos and colleagues have submitted a paper entitled “The Impact of EndoVAC in Addressing Post-Esophagectomy Anastomotic Leak in Esophageal Cancer Management” for consideration for publication in the Journal of Clinical Medicine.
This is a well written review and analysis of the use of the end therapy for treatment of anastomotic leak post esophagectomy. It is timely, helpful, and worthwhile. The following comments are intended to help improve the impact and relevance of the paper.
The four sections of section 3 are well written and form the major contributions of the paper. One suggestion for the authors to consider is to include is a fifth section, namely on, suggestions for ongoing research regarding the EndoVAC. A discussion regarding remaining unsolved problems, areas to focus for future investigations, and other commentary regarding future academic work would benefit the paper.
There is no mention of the combination therapy offered by the 'EndoVAC stent', which is a newly approved tool in Europe for management of anastomotic leak. The authors discussed the use of the EndoVAC as compared to use of a stent, but the discussion would benefit from discussion regarding their combination.
Suggest to be more concise with language. For example, the abstract can be truncated without loss of content – for example these sentences in the abstract could be reduced. “Anastomotic leakage (AL) stands as one of the most serious and feared complications following gastrointestinal surgery. Among upper gastrointestinal surgeries, esophagectomy emerges as particularly challenging and is often complicated by this type of leakage. This is especially true for patients with esophagogastric malignancies, who face an increased risk of surgical complications. This elevated risk can be attributed to the frequent use of neoadjuvant treatment approaches, including chemotherapy and radiotherapy, which may compromise tissue integrity and healing processes.” The rest of the abstract provides a clear description of what the paper is about, namely to ”analyze the role and effectiveness of EndoVAC in the therapeutic management of AL”.
Suggest to be more precise with your language. For example, the first sentence of section 2 states that “As mentioned above, AL is the most dreadful complication after esophagectomy”. The word dreadful is of course subjective and difficult to define. Gastric conduit necrosis and fistulas into the airway are arguably more “dreadful”, yet thankfully far more rare. Anastomotic leak is a profoundly impactful adverse event, and based on its incidence and impact, it may be the most negatively impactful post esophagectomy. Suggest to rephrase.
The authors might discuss the health economic implications of both anastomotic leak (i.e. cost of the adverse event) as well as the costs of the use of the EndoVAC, as it requires multiple endoscopies (i.e. costs of the therapy). Suggest to include this health economic discussion. If that is outside the scope of the paper, it could be highlighted in the suggested section on future research.
Author Response
Comment 1: The four sections of Section 3 are well-written and form the major contributions of the paper. One suggestion for the authors to consider is to include a fifth section on suggestions for ongoing research regarding the EndoVAC. A discussion regarding remaining unsolved problems, areas to focus for future investigations, and other commentary regarding future academic work would benefit the paper.
Response 1: Thank you for your valuable comments which significantly improved our manuscript. We appreciate this suggestion and have added a fifth section focused on future research directions for EndoVAC. This new section discusses unresolved issues, emerging areas for research, and opportunities for interdisciplinary studies to improve EVT application and outcomes.
Comment 2: There is no mention of the combination therapy offered by the 'EndoVAC stent,' a newly approved tool in Europe for managing anastomotic leak. The authors discussed the use of the EndoVAC as compared to the use of a stent, but the discussion would benefit from including this combination.
Response 2: Thank you for your feedback. We incorporated a discussion on the EndoVAC stent. This section explores how combining EndoVAC and stenting may benefit complex cases, offering a hybrid approach that leverages the strengths of both therapies.
Comment 3: Suggest to be more concise with language. For example, the abstract can be truncated without loss of content.
Response 3: Thank you for this recommendation. We have revised the abstract and key sections for conciseness.
Comment 4: Suggest to be more precise with your language. For example, the word "dreadful" is subjective. Anastomotic leak is a profoundly impactful adverse event, and based on its incidence and impact, it may be the most negatively impactful post-esophagectomy. Suggest rephrasing.
Response 4: Thank you for your comment. Terms like "dreadful" have been replaced with objective descriptors
Comment 5: The authors might discuss the health economic implications of both anastomotic leak (i.e., the cost of the adverse event) as well as the costs of using EndoVAC, as it requires multiple endoscopies.
Response 5: Thank you for your input. We added a paragraph discussing the economic implications of AL and EndoVAC therapy. Additionally, we suggest future studies on the cost-benefit analysis of EVT to provide further insights.
Reviewer 3 Report
Comments and Suggestions for Authors
Thank you for the opportunity to review this work. I understand that this is a narrative review, but I would like to ask the authors why they did not choose to conduct a systematic review. Additionally, why did the authors decide against synthesizing the data, for example, by creating a meta-analysis?
Comments
At the end of the introduction, please clearly state the purpose of the work.
Please justify the alignment of the introduction.
Line 114 – ‘(p=0.006’ and throughout the entire work. The authors report statistical analysis values multiple times. In my opinion, these are incorrect and confusing; please simply indicate whether the result was statistically significant or not. Providing only "p" values without additional information, such as effect size, may lead to interpretive errors.
Please check abbreviations, such as ‘SEMS,’ which is expanded twice (Line 236 and Line 311). Please verify all abbreviations.
References – please standardize formatting, as some entries are capitalized (e.g., Reference 47), while others lack page numbers or bibliometric information (e.g., Reference 6). DOI numbers are also missing. The references appear as though the authors did not use a bibliography manager.
Author Response
Comment 1: I understand that this is a narrative review, but I would like to ask the authors why they did not choose to conduct a systematic review or meta-analysis.
Response 1: Thank you for your comment. We opted for a narrative review due to the heterogeneity in available studies and the diverse methodologies across EVT literature, which limit the feasibility of a meta-analysis.
Comment 2: At the end of the introduction, please clearly state the purpose of the work.
Response 2: We appreciate this suggestion and have revised the end of the introduction to clearly state the purpose of the review.
Comment 3: The authors report statistical analysis values multiple times, which may be confusing; please simplify by indicating only whether the result was statistically significant or not.
Response 3: Thank you for your comment. We have simplified the reporting of statistical values and included effect sizes when available to provide context.
Comment 4: Please check abbreviations, such as 'SEMS,' which is expanded twice. Verify all abbreviations.
Response 4: Thank you for your recommendation. We reviewed and standardized all abbreviations throughout the manuscript to ensure each term is only defined once on first use.
Comment 5: References – please standardize formatting, as some entries are capitalized inconsistently, while others lack page numbers or DOI numbers.
Response 5: Thank you for your feedback. We used EndNote reference manager using the standardized MDPI template to ensure the uniformity of references.